# SPHEREFACE2: BINARY CLASSIFICATION IS ALL YOU NEED FOR DEEP FACE RECOGNITION

**Yandong Wen[1],\***, **Weiyang Liu[2,3],\***, **Adrian Weller[2,4]**, **Bhiksha Raj[1]**, **Rita Singh[1]**
[1]Carnegie Mellon University  [2]University of Cambridge  [3]MPI for Intelligent Systems  [4]Alan Turing Institute

## ABSTRACT

State-of-the-art deep face recognition methods are mostly trained with a softmax-based multi-class classification framework. Despite being popular and effective, these methods still have a few shortcomings that limit empirical performance. In this paper, we start by identifying the discrepancy between training and evaluation in the existing multi-class classification framework and then discuss the potential limitations caused by the "competitive" nature of softmax normalization. Motivated by these limitations, we propose a novel binary classification training framework, termed *SphereFace2*. In contrast to existing methods, SphereFace2 circumvents the softmax normalization, as well as the corresponding closed-set assumption. This effectively bridges the gap between training and evaluation, enabling the representations to be improved individually by each binary classification task. Besides designing a specific well-performing loss function, we summarize a few general principles for this "one-vs-all" binary classification framework so that it can outperform current competitive methods. Our experiments on popular benchmarks demonstrate that SphereFace2 can consistently outperform state-of-the-art deep face recognition methods. The code is available at OpenSphere.

## 1 INTRODUCTION

Recent years have witnessed the tremendous success of deep face recognition (FR), largely owing to rapid development in training objectives [4, 16–18, 31, 32, 37–39, 42, 43]. Current deep FR methods are typically based on a multi-class learning objective, *e.g.*, softmax cross-entropy loss [4, 17, 18, 27, 37–39]. Despite its empirical effectiveness, there is an obvious discrepancy between such a multi-class classification training and open-set pair-wise testing, as shown in Fig. 1. In contrast to multi-class classification, pair-wise verification is a binary problem where we need to determine whether a pair of face images belongs to the same person or not. This significant discrepancy may cause the training target to deviate from the underlying FR task and therefore limit the performance. This problem has also been noticed in [29, 31], but they still try to address it under the multi-class (or triplet) learning framework which still fundamentally differs from pair-wise verification in testing. On the other hand, multi-class classification training assumes a closed-set environment where all the training data must belong to one of the known identities, which is also different from open-set testing.

In order to address these limitations, we propose a novel deep face recognition framework completely based on binary classification. SphereFace [17] is one of the earliest works that explicitly performs multi-class classification on the hypersphere (*i.e.*, angular space) in deep FR. In light of this, we name our framework *SphereFace2* because it exclusively performs binary classi-

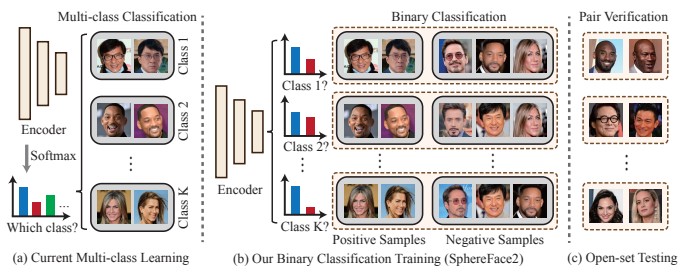

Figure 1: Comparison between current multi-class classification training in deep face recognition and our binary classification training.

fication (hence the "2") on the hypersphere. Unlike multi-class classification training, SphereFace2 effectively bridges the gap between training and testing, because both training and testing perform pair-wise comparisons. Moreover, SphereFace2 alleviates the closed-set assumption and views the training as an open-set learning problem. For example, training samples that do not belong to any class are still useful and can serve as negative samples in SphereFace2, while they cannot be used

---

\*Yandong Wen and Weiyang Liu are co-first authors and contributed equally to this work.

at all in the multi-class classification framework. Specifically, given $K$ classes in the training set, SphereFace2 constructs $K$ binary classification objectives where data from the target class are viewed as positive samples and data from the remaining classes are negative samples. The spirit is similar to "one-vs-all" classification [28]. An intuitive illustration comparing typical multi-class training, our proposed SphereFace2, and open-set evaluation is given in Fig. 1.

In multi-class classification training, the softmax (cross-entropy) loss introduces competition among different classes, because the softmax normalization requires all the class logits are summed to one. Therefore, increasing the confidence of one class will necessarily decrease the confidence of some other classes, which can easily lead to over-confident predictions [6]. In contrast to the competitive nature in all the softmax losses, our SphereFace2 encourages each binary classification task to individually improve the discriminativeness of the representation without any competition.

The significance of our method can also be understood from a different perspective. Depending on how samples interact with each other in training, deep FR methods can be categorized into triplet-based learning and pair-based learning. Triplet-based learning simultaneously utilizes an anchor, positive samples and negative samples in its objective, while pair-based learning uses either positive pairs or negative pairs in one shot. Based on whether a proxy is used to represent a set of samples, both triplet-based and pair-based methods can learn with or without proxies. A categorization of deep FR methods is given in Table 1. Proxy-free methods usually require expansive pair/triplet mining and most of them [8, 23, 26, 31, 33, 52] use a hybrid loss that includes the standard softmax loss. Typical examples of triplet-based learning with proxy include the softmax loss and its popular variants [4, 17, 37–39] where a classifier is a class proxy and the learning involves an anchor (*i.e.*, the deep feature

|  | w/o Proxy | w/ Proxy |
|---|---|---|
| Triplet | FaceNet [29]
VGGFace [26]
Triplet [23]
MTER [52] | DeepID [32]
DeepFace [35]
DeepID2* [31]
DeepID2+* [33]
SphereFace [17]
NormFace [38]
CosFace [37, 39]
ArcFace [4]
Ring Loss [51]
CurricularFace [10]
Circle Loss [34] |
| Pair | Siamese [3]
DeepID2* [31]
DeepID2+* [33]
Cont. CNN [8] | **SphereFace2** |

Table 1: Overview of deep FR. * denotes that the method uses a hybrid loss that combines a multi-class softmax loss and a contrastive loss.

$\boldsymbol{x}$), the positive proxy (*i.e.*, the target classifier $\boldsymbol{W}_y$) and the negative proxies (*i.e.*, the other classifiers $\boldsymbol{W}_j, j \neq y$). Most popular deep FR methods use proxies by default, since they greatly speed up training and improve data-efficiency (especially on large-scale datasets). Distinct from previous deep FR, we believe SphereFace2 is the first work to adopt a *pair-wise* learning paradigm with proxies.

An outstanding difference between triplet-based and pair-based learning is the usage of a universal threshold. In Fig. 2, we show that triplet-based learning compares the similarity scores between different pairs, while pair-based learning compares a similarity score and a universal threshold. As a pair-based method, SphereFace2 too optimizes the difference between similarity

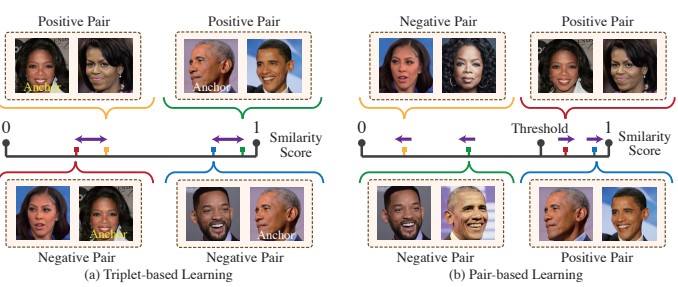

Figure 2: Comparison between triplet-based and pair-based learning. Purple arrows denote optimization directions. Triplet-based learning compares different similarity scores, while pair-based learning compares similarity score and a threshold.

scores and a universal threshold. By learning the threshold to distinguish positive and negative pairs during training, it naturally also generalizes to open-set FR.

In our framework, we cast a principled view on designing loss functions by systematically discussing the importance of positive/negative sample balance, easy/hard sample mining and angular margin, and combine these ingredients to propose an effective loss function for SphereFace2. In fact, most popular variants of the softmax loss [4, 37–39] also perform easy/hard sample mining and incorporate angular margin. Moreover, we observe that the distribution of similarity scores from positive pairs is quite different from that of negative pairs, in that it exhibits larger variance, which makes the threshold hard to determine. To address this, we propose a novel similarity adjustment method that can adjust the distribution of pair similarity and effectively improve the generalizability.

The flexibility of binary classifications in SphereFace2 brings a few additional advantages. First, in contrast to the classifier layer with the softmax loss which is highly non-trivial to parallelize [1], the

classifier layer in SphereFace2 can be naturally parallelized across different GPUs. While the gradient update in the softmax loss needs to involve all the classifiers, the gradient update for classifiers in SphereFace2 is independent and fully decoupled from each other. The gradient with respect to each classifier only depends on the feature $x$ and the binary classifier itself. Therefore, SphereFace2 naturally addresses the problem of the softmax loss in training on a large number of face identities [1]. Second, the pair-wise labels used in SphereFace2 are in fact a weaker form of supervision than the class-wise labels used in all the softmax-based losses. This property is closely connected to the open-set assumption and also makes SphereFace2 less sensitive to class-wise label noise. For example, SphereFace2 does not require the class-wise labels of the negative samples in each binary classification. Instead, as long as we know that the identities of negative samples are different from the positive samples, SphereFace2 will still work well. Our experiments also empirically verify the robustness of SphereFace2 against class-wise label noise. Our major contributions are:

- SphereFace2 constructs a novel binary classification framework for deep FR. To our knowledge, SphereFace2 is the first work in deep FR to adopt a pair-wise learning paradigm with proxies. We further summarize a series of general principles for designing a good loss function for SphereFace2.
- The decoupling of the binary classifications leads to a natural parallelization for the classifier layer, making SphereFace2 more scalable than softmax-based losses.
- The pair-wise labels used in SphereFace2 are a weaker supervision than the class-wise labels used in the softmax loss, yielding better robustness to class-wise label noise.

**Related Work**. Deep FR methods are either proxy-based or proxy-free, as shown in Table 1. Proxy-free learning includes contrastive loss [3, 7, 31] and triplet loss [29]. Both losses directly optimize the similarity between samples, so they are highly unstable and data-inefficient. Proxy-free learning is more widely used in deep metric learning [25, 30, 36, 40, 41, 45]. Proxy-based learning uses a set of proxies to represent different groups of samples and usually works better for large-scale datasets. Typical examples include the softmax loss and its variants [4, 17, 32, 34, 35, 37–39] where each proxy is used to represent a class. SphereFace [17] suggests that large-margin classification is more aligned with open-set FR, and proposes to learn features with large angular margin. CosFace [37, 39] and ArcFace [4] introduce alternative ways to learn large-margin features with improved stability. Although large-margin classification brings the training target closer to the task of open-set FR, the discrepancy still exists and it is unclear how much the margin should be to close the gap. Moreover, a large margin inevitably leads to training instability. In contrast, SphereFace2 naturally avoids these problems and aligns the training target with open-set FR by adopting pair-wise learning with proxies.

## 2 THE SPHEREFACE2 FRAMEWORK

### 2.1 OVERVIEW AND PRELIMINARIES

The goal of SphereFace2 is to align the training target with open-set verification so that our training is more effective in improving open-set generalization in deep FR. To this end, SphereFace2 explicitly incorporates pair-wise comparison into training by constructing $K$ binary classification tasks ($K$ is the number of identities in the training set). The core of SphereFace2 is the binary classification reformulation of the training target. In the $i$-th binary classification, we construct the positive samples with the face images from the $i$-th class and the negative samples with face images from other classes. Specifically, we denote the weights of the $i$-th binary classifier by $W_i$, the deep feature by $x$ and its corresponding label by $y$. A naive loss formulation is

$$L_f = \log\left(1 + \exp(-W_y^\top x - b_y)\right) + \sum_{i \neq y} \log\left(1 + \exp(W_i^\top x + b_i)\right)$$

which is a combination of $K$ standard binary logistic regression losses. Instead of performing binary classification in a unconstrained space, we perform binary classification on the unit hypersphere by normalizing both classifiers $W_i, \forall i$ and feature $x$. The loss function now becomes

$$L_s = \log\left(1 + \exp(-\cos(\theta_y))\right) + \sum_{i \neq y} \log\left(1 + \exp(\cos(\theta_i))\right) \tag{1}$$

where $\theta_i$ is the angle between the $i$-th binary classifier $W_i$ and the sample $x$. The biases $b_i, \forall i$ are usually removed in common practice [4, 17, 19, 37, 39], since they are learned for a closed set and cannot generalize to unknown classes. However, we actually find them very useful in SphereFace2, as will be discussed later. For now, we temporarily remove them for notational convenience. One of the unique advantages of such a parameterization for binary classifiers is that it constructs the $i$-th class positive proxy with $W_i$ and the negative proxy with $-W_i$. Depending on the label, the

training will minimize the angle between $x$ and $W_i$ or between $x$ and $-W_i$ in order to minimize the loss. This parameterization of positive and negative proxies immediately guarantees minimum hyperspherical energy [12–15] that has been shown to effectively benefit generalization. Moreover, our parameterization in SphereFace2 has the same number of parameters for the classifier layer as the previous multi-class training, and does not introduce extra overhead. However, naively minimizing the loss in Eq. (1) will not give satisfactory results. Therefore, we explore in the next subsection how to find a desirable loss function that works well in the SphereFace2 framework.

## 2.2 A PRINCIPLED VIEW ON LOSS FUNCTION

We emphasize that the exact form of our loss function is in fact not crucial, and the core of SphereFace2 is the spirit of binary classification (*i.e.*, pair-wise learning with proxies). Following such a spirit, there are likely many alternative losses that work as well as ours. Besides proposing a specific loss function, we summarize our reasoning for designing a good loss function via a few general principles.

**Positive/negative sample balance**. The first problem in SphereFace2 is how to balance the positive and negative samples. In fact, balancing positive/negative samples has also been considered in triplet loss, contrastive loss and softmax-based losses. Contrastive loss achieves the positive/negative sample balance by selecting the pairs. Based on [34], triplet-based methods including both triplet loss and softmax-based losses can naturally achieve positive/negative sample balance, since these losses require the presentation of a balanced number of both positive and negative samples.

From Eq. (1), the gradients from positive samples and negative samples are highly imbalanced because only one out of $K$ terms computes the gradient for positive samples. A simple yet effective remedy is to introduce a weighting factor $\lambda$ to balance gradients for positive and negative samples:

$$L_b = \lambda \log \left(1 + \exp(-\cos(\theta_y))\right) + (1-\lambda) \sum_{i \neq y} \log \left(1 + \exp(\cos(\theta_i))\right)$$

where $\lambda \in [0, 1]$ is a hyperparameter that determines the balance between positive and negative samples. One of the simplest ways to set $\lambda$ is based on the fraction of the loss terms, *i.e.*, $\lambda = \frac{K-1}{K}$ when there are $K$ classes in total.

**Easy/hard sample mining**. Another crucial criterion for a good loss function is the strategy for mining easy/hard samples, since it is closely related to the convergence speed and quality. It is commonly perceived that softmax-based losses are free of easy/hard sample mining, unlike triplet and contrastive losses. However, this is inaccurate. We construct

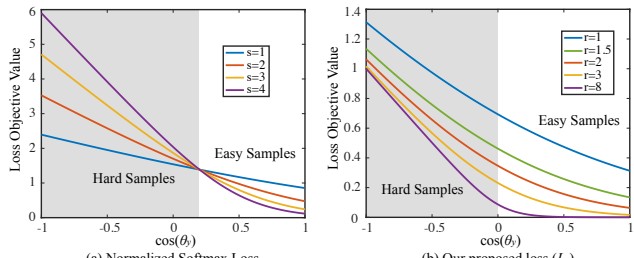

Figure 3: Loss objective value under different target cosine values.

a simple example to illustrate how the softmax-based loss mines easy/hard samples. We assume a set of cosine logits from four classes is $[\cos(\theta_1), \cos(\theta_2), \cos(\theta_3), \cos(\theta_4)]$ and the target class is $y = 1$. Then we also compute the $s$-normalized softmax loss $L_n = -\log(\frac{\exp(s \cdot \cos(\theta_y))}{\sum_i \exp(s \cdot \cos(\theta_i))})$ by fixing $\cos(\theta_i) = 0.2, i \neq y$ and varying $\cos(\theta_y)$ from $-1$ to $1$. From the results shown in Fig. 3(a), we can observe that as the scale $s$ increases, the loss for hard samples will be higher and also more sensitive than the loss for easy samples. This makes the neural network focus on optimizing the angles of hard samples and therefore can improve convergence and generalization. The scaling strategy is widely used as a common practice in most softmax-based margin losses [4, 34, 37–39], playing an implicit role of easy/hard sample mining. For the standard softmax cross-entropy loss, easy/hard sample mining is dynamically realized by the norm of features and classifiers.

In our framework, we need a similar strategy to mine easy/hard samples. Inspired by the rescaled softplus function [5], we use an extra hyperparameter $r$ to adjust the curvature of the loss function $L_b$:

$$L_e = \frac{\lambda}{r} \cdot \log \left(1 + \exp(-r \cdot \cos(\theta_y))\right) + \frac{1-\lambda}{r} \cdot \sum_{i \neq y} \log \left(1 + \exp(r \cdot \cos(\theta_i))\right) \tag{2}$$

where larger $r$ implies stronger focus on hard samples. We consider an example of one binary classification and $L_e$ becomes $\frac{1}{r} \cdot \log \left(1 + \exp(-r \cdot \cos(\theta_y))\right)$ when $\lambda = 1$. We then plot how the loss value changes by varying $\cos(\theta_y)$ from $-1$ to $1$ under different $r$ in Fig. 3(b). We observe that when $r$ becomes larger, the loss for easy samples gets closer to zero while the loss for hard samples remains large. Therefore, $r$ can help to mine and reweight easy/hard samples during training.

**Angular margin.** Learning deep features with large angular margin is arguably one of the most effective criteria to achieve good generalizability on open-set face recognition. SphereFace [17] introduced angular margin to deep face recognition by considering a multiplicative margin. CosFace [37, 39] and ArcFace [4] further considered an additive angu-

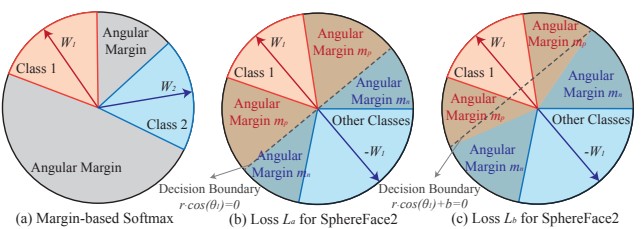

Figure 4: Intuitive comparison of angular margin in different losses.

lar margin which makes the loss function easier to train. In light of these works, we introduce a novel two-sided angular margin with two adjustable parameters to the SphereFace2 framework:

$$L_a = \frac{\lambda}{r} \cdot \log\left(1 + \exp(-r \cdot (\cos(\theta_y) - m_p))\right) + \frac{1-\lambda}{r} \cdot \sum_{i \neq y} \log\left(1 + \exp(r \cdot (\cos(\theta_i) + m_n))\right) \quad (3)$$

where $m_p$ controls the size of the margin for positive samples and $m_n$ controls the size of the margin for negative samples. Larger $m_p$ and $m_n$ lead to larger additive angular margin, sharing similar spirits to CosineFace [37, 39]. Our framework is agnostic to different forms of angular margin and all types of angular margin are applicable here. We also apply ArcFace-type margin [4] and multiplicative margin [16, 17] to SphereFace2 and obtain promising results. Details are in Appendix F.

However, quite different from the angular margin in the softmax-based losses, the angular margin in Eq. 3 has a universal confidence threshold 0 (*i.e.*, $\cos(\theta_i) = 0$). Both angular margins $m_p$ and $m_n$ are introduced with respect to this decision boundary $\cos(\theta_i) = 0$, see Fig. 4(b). This property has both advantages and disadvantages. One of the unique advantages is that our angular margin for each class is added based on a universally consistent confidence threshold and does not depend on the other classifiers, while the angular margin in softmax-based losses will be largely affected by the neighbor classifiers. However, it is extremely challenging to achieve the universal threshold 0, which results in training difficulty and instability. To improve training stability, the bias term that has long been forgotten in softmax-based losses comes to the rescue. We combine the biases back:

$$L_b = \frac{\lambda}{r} \cdot \log\left(1 + \exp(-r \cdot (\cos(\theta_y) - m_p) - b_y)\right) + \frac{1-\lambda}{r} \cdot \sum_{i \neq y} \log\left(1 + \exp(r \cdot (\cos(\theta_i) + m_n) + b_i)\right) \quad (4)$$

where $b_i$ denotes the bias term for the binary classifier of the $i$-th class. Since the class-specific bias is not useful in the open-set testing, we will simply use the same bias $b$ for all the classes. The bias $b$ now becomes the universal confidence threshold for all the binary classifications and the baseline decision boundary becomes $r \cdot \cos(\theta_y) + b = 0$ instead of $r \cdot \cos(\theta_y) = 0$, making the training more stable and flexible. The final decision boundary is $r \cdot (\cos(\theta_y) - m_p) + b = 0$ for the positive samples and $r \cdot (\cos(\theta_y) + m_n) + b = 0$ for the negative samples. More importantly, the threshold $b$ is still universal and consistent for each class. An illustration of Eq. 4 is given in Fig. 4. Alternatively, we can interpret Eq. 4 as a learnable angular margin where the confidence threshold is still 0. Then we can view $m_p - \frac{b}{r}$ as the positive margin and $m_n + \frac{b}{r}$ as the negative margin. Because $b$ is learnable, we only need to focus on $m_p + m_n$ which yields only one effective hyperparameter. For convenience, we simply let $m_p = m_n = m$ and only need to tune $m$ in practice.

**Similarity adjustment.** In Fig. 5(a), we observe a large inconsistency between positive and negative pairs in the distribution of cosine similarity. The similarity distribution of negative pairs has smaller variance and is more concentrated, while the positive pairs exhibit much larger variation in similarity score. The distributional discrepancy between positive and negative pairs leads to a large overlap of

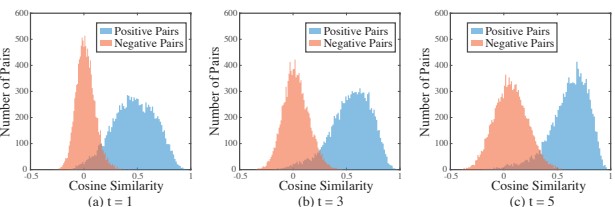

Figure 5: Similarity score distribution of positive and negative pairs trained with different $t$. The evaluation pairs are combined from 4 sets: LFW [9], Age-DB30 [24], CA-LFW [49] and CP-LFW [50].

similarity scores between them, making it difficult to give a clear threshold to separate the positive and negative pairs. This is harmful to generalization. Fig. 5(a) also empirically shows the similarity scores mostly lie in the range of $[-0.2, 1]$. These observations motivate us to (1) reduce the overlap of the similarity scores between positive and negative pairs, and (2) increase the empirical dynamic range of the similarity score such that the pair similarity can be distributed in a larger space.

To this end, we propose a novel similarity adjustment method. Our basic idea is to construct a monotonic decreasing function $g(z)$ where $z \in [-1, 1]$ and $g(z) \in [-1, 1]$ and then use $g(\cos(\theta))$ instead of the original $\cos(\theta)$ to adjust the mapping from angle to similarity score during training. Considering that the originally learned $\cos(\theta)$ mostly lies in the range of $[-0.2, 1]$, we require $g(z)$ to map $[-0.2, 1]$ to a larger range (*e.g.*, $[-0.9, 1]$), so that if $\cos(\theta)$ is learned similarly as before and still gives the empirical dynamic range of $[-0.2, 1]$, we can end up with $g(\cos(\theta))$ whose empirical dynamic range becomes $[-0.9, 1]$. Specifically, $g(z)$ is parameterized as $g(z) = 2\left(\frac{z+1}{2}\right)^t - 1$

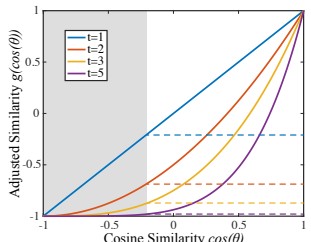

Figure 6: $g(\cos(\theta))$ of different $t$.

where we typically use $z = \cos(\theta) \in [-1, 1]$. In practice, we simply replace the original cosine similarity with $g(\cos(\theta))$ during training. $t$ controls the strength of similarity adjustment. When $t = 1$, $g(\cos(\theta))$ reduces to the standard cosine similarity. In Fig. 5, we show that the similarity distribution can be modified by increasing the parameter $t$. As $t$ increases, the overlap between the positive and negative pairs is reduced and their similarity distributions also become more separable. Moreover, the empirical dynamic range of the similarity score is also approximately increased from $[-0.2, 1]$ to $[-0.4, 1]$. The empirical results validate the effectiveness of the proposed similarity adjustment.

**Final loss function**. After combining all the principles and simplifying hyperparameters, we have

$$L = \frac{\lambda}{r} \cdot \log\left(1 + \exp\left(-r \cdot (g(\cos(\theta_y)) - m) - b\right)\right) + \frac{1 - \lambda}{r} \cdot \sum_{i \neq y} \log\left(1 + \exp\left(r \cdot (g(\cos(\theta_i)) + m) + b\right)\right) \quad (5)$$

where $g(\cdot)$ has a hyperparameter $t$. In total, there are four hyperparameters $\lambda$, $r$, $m$ and $t$. Each has a unique geometric interpretation, making them easy to tune. Following our design principles, there could be many potential well-performing loss functions that share similar properties to the proposed one. Our framework opens up new possibilities to advance deep face recognition.

### 2.3 Geometric Interpretation

This subsection provides a comprehensive discussion and visualization to justify our designs and explain the geometric meaning of each hyperparameter. By design, $r$ is the radius of the hypersphere where all the learned features live and is also the magnitude of the features. The bias $b$ for the $i$-th class moves the baseline decision boundary along the direction of its classifier $\boldsymbol{W}_i$. The parameter $m$ controls the size of the induced angular margin. We set the output feature dimension as 2 and plot the 2D features trained by SphereFace2 with different margin $m$ in Fig. 7. The visualization empirically verifies the following arguments.

**The bias $b$ moves the decision boundary**. From Fig. 7, we can observe that the bias $b$ can be effectively learned to move the decision boundary along the classifier direction and lead to a new universal confidence $-b$ for all the classes. The bias $b$ makes the training easier and more stable while still preserving the unique property that all classes share the same confidence for classification. Compared to other deep FR methods, the universal confidence in SphereFace2 can help to learn a consistent positive/negative pair separation and explicitly encourage a unique and consistent verification threshold during training.

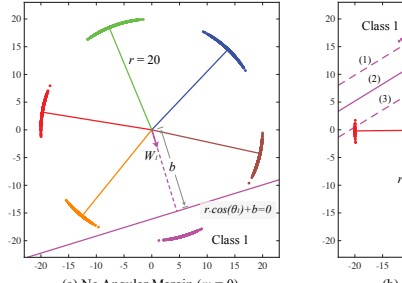
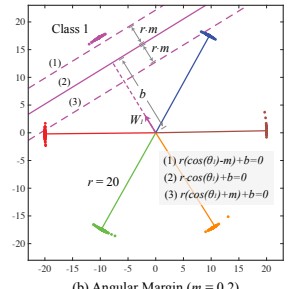

Figure 7: 2D deep feature learned by SphereFace2. We construct a small dataset consisting of 6 face identities from VGGFace2 [2]. Dots with the same color represent samples from the same face identity.

**$m$, $r$ control the angular margin**. Fig. 7 visualizes the baseline decision boundary (denoted as (2) in Fig. 7(b)) and the decision boundary for the positive/negative samples (denoted as (1)/(3) in Fig. 7(b)). The distance between the positive and negative decision boundary is $2rm$, producing an effective angular margin. The results show that the empirical margin matches our expected size well.

**Larger $m$ leads to larger angular margin**. From Fig. 7, we can empirically compare the deep features learned with $m = 0$ and $m = 0.2$ and verify that larger $m$ indeed incorporates larger angular margin between different classes. Large inter-class separability is also encouraged.

We also visualize 3D deep features and the decision planes for one class with $r = 30$ and $m = 0.2$ in Fig. 8. We can observe that samples from each class are well separated with large angular margins, which is consistent with the 2D case. The empirical angular margin also perfectly matches the induced one (*i.e.*, the distance between the positive and negative plane). The results further verify our empirical conclusions drawn from the 2D visualization.

Figure 8: 3D features.

### 2.4 Efficient Model Parallelization on GPUs

As it becomes increasingly important for deep FR methods to train on large-scale datasets with million-level identities, a bottleneck is the storage of the classifier layer since its space complexity grows linearly with the number of identities. A common solution is to distribute the storage of the classifiers $W_i, \forall i$ and their gradient computations to multiple GPUs. Then each GPU only needs to compute the logits for a subset of the classes. However, the normalization in softmax-based losses inevitably requires some data communication overhead across different GPUs, resulting in less efficient parallelization. In contrast, the gradient computations in SphereFace2 are class-independent and can be performed locally within one GPU. Thus no communication cost is needed. The decoupling among different classifiers makes SphereFace2 suitable for multi-GPU model parallelization.

Specifically, the softmax normalization in the softmax loss involves the computation of all the classifiers, so computing the gradient *w.r.t.* any classifier will still require the weights of all the other classifiers, which introduces communication overhead across GPUs. In contrast, the loss for SphereFace2 (Eq. (5)) can be rewritten as $L = \sum_{i=1}^{K} f_i(\boldsymbol{W}_i, \boldsymbol{x})$ where $f_i(\cdot, \cdot)$ is some differentiable function. To compute $\frac{\partial L}{\partial \boldsymbol{W}_i}$, we only need to compute $\frac{\partial f_i(\boldsymbol{W}_i, \boldsymbol{x})}{\boldsymbol{W}_i}$ which does not involve any other classifiers. Therefore, this gradient computation can be performed locally and does not require any communication overhead. Appendix E compares the gradient of SphereFace2 and the softmax loss.

## 3 Experiments and Results

**Preprocessing**. Each face image is cropped based on the 5 face landmarks detected by MTCNN [48] using a similarity transformation. The cropped image is of size $112 \times 112$. Each RGB pixel ([0, 255]) is normalized to $[-1, 1]$. We put the details of training and validation datasets in Appendix A.

**CNNs**. We adopt the same CNNs from [17, 39] for fair comparison. We use 20-layer CNNs in ablations and 64-layer CNNs for the comparison to existing state-of-the-art methods.

**Training and Testing**. We use SGD with momentum 0.9 by default. We adopt VGGFace2 [2] as the same training set for all the methods. VGGFace2 contains 3.1M images from 8.6K identities, as shown in Table 9. The training faces are horizontally flipped for data augmentation. We strictly follow the specific protocol provided in each dataset for evaluations. Given a face image, we extract a 512D embedding. The final score is computed by the cosine distance of two embeddings. The nearest neighbor classifier and thresholding are used for face identification and verification, respectively.

### 3.1 Ablation Study and Exploratory Experiments

We perform an ablation study on four validation sets: LFW, AgeDB-30, CA-LFW, and CP-LFW. The statistics of these datasets are summarized in Table 9. Following the provided evaluation protocols, we report 1:1 verification accuracy of 6,000 pairs (3,000 positive and 3,000 negative pairs) for each dataset. In addition, we combine these datasets and compute the overall verification accuracy, which serves as a more accurate metric to evaluate the models.

**Design Principles**. We start with ablations on the designing principles of the loss function. As shown in Table 2, it is quite effective to improve the performance following the principles to design a loss function. Specifically, the naive binary classification (Eq. 1) fails to converge, since the learning objective and

| PN | EH | AM | SA | LFW | AgeDB-30 | CA-LFW | CP-LFW | Combined |
|----|----|----|----|-----|----------|--------|--------|----------|
| ✗ | ✗ | ✗ | ✗ | 93.60 | 71.67 | 68.40 | 74.07 | 74.49 |
| ✓ | ✗ | ✗ | ✗ | 95.37 | 72.90 | 72.93 | 76.90 | 78.03 |
| ✓ | ✓ | ✗ | ✗ | 98.60 | 88.10 | 89.98 | 85.23 | 89.65 |
| ✓ | ✓ | ✓ | ✗ | **99.62** | 92.82 | 93.07 | 90.85 | 93.87 |
| ✓ | ✓ | ✓ | ✓ | 99.50 | **93.68** | **93.47** | **91.07** | **94.28** |

Table 2: Ablations of designing principles. PN, EH, AM and SA are the abbreviations for positive/negative sample balance, easy/hard sample mining, angular margin, and similarity adjustment (%).

the gradients are both dominated by the negative pairs. To address this, we set $\lambda$ to $\frac{K-1}{K}$ and report the results in the first row of Table 2. As can be seen, while the model is trainable and starts to converge, the results show significant room for improvement. After some tuning of the hyperparameter $\lambda$,

SphereFace2 can achieve 78.03% accuracy on the combined dataset. Then we gradually incorporate the hard sample mining and angular margin into SphereFace2, improving the verification accuracy from 78.03% to 89.65%, and then to 93.87% With similarity adjustment, SphereFace2 yields 94.28% accuracy on the combined dataset. Table 2 clearly shows the effectiveness of each ingredient.

**Hyperparameters.** There are four hyperparameters ($\lambda$, $r$, $m$, and $t$) in SphereFace2. Since $r$ and $m$ have been extensively explored in [4, 37–39], we follow previous practice to fix $r$ and $m$ to 30 and 0.4 respectively. Our experiments mainly focus on analyzing the effect of $\lambda$ and $t$.

| $\lambda$ | LFW | AgeDB-30 | CA-LFW | CP-LFW | Combined |
|---|---|---|---|---|---|
| 0.3 | 99.38 | 91.38 | 92.93 | 88.88 | 92.72 |
| 0.4 | 99.42 | 92.30 | 92.85 | 89.97 | 93.38 |
| 0.5 | 99.55 | 92.77 | 93.32 | 90.20 | 93.75 |
| 0.6 | 99.48 | 92.67 | **93.40** | 90.10 | 93.63 |
| 0.7 | 99.58 | 92.63 | 93.30 | 90.33 | 93.73 |
| 0.8 | **99.62** | **92.81** | 93.07 | **90.85** | **93.87** |
| 0.9 | 99.50 | 92.57 | 92.82 | 90.53 | 93.62 |
| 0.99 | 99.53 | 90.37 | 91.68 | 89.33 | 92.31 |

Table 3: Effect of different $\lambda$. We fix $t = 1$ and explore how the model performs with different $\lambda$s. Results are in %.

From Table 3, the performance of SphereFace2 remains stable for $\lambda$ from 0.5 to 0.9, where a good balance for the positive and negative pairs is achieved. Then we evaluate different $t$ for $\lambda = [0.6, 0.7, 0.8]$ and report the results in Table 4. As can be seen from the results, adjusting the similarity scores further boosts model accuracy. The effect of different $t$ is also illustrated in Fig. 5. More detailed ablation studies are included in Appendix C and D.

| $\lambda$ | $t$ | LFW | AgeDB-30 | CA-LFW | CP-LFW | Combined |
|---|---|---|---|---|---|---|
| 0.6 | 2 | 99.53 | 93.30 | 93.37 | 90.65 | 94.02 |
| 0.6 | 3 | 99.48 | **93.80** | **93.53** | 91.08 | **94.28** |
| 0.7 | 2 | **99.62** | 93.22 | 93.35 | 91.02 | 94.05 |
| 0.7 | 3 | 99.50 | 93.68 | 93.47 | 91.07 | **94.28** |
| 0.8 | 2 | 99.57 | 93.55 | 93.28 | 90.72 | 94.03 |
| 0.8 | 3 | **99.62** | 93.58 | 93.38 | **91.12** | 94.23 |

Table 4: Effect of different $t$. We explore different $t$s for several besting performing $\lambda$s. Results are in % and higher is better.

**State-of-art loss functions**. We compare SphereFace2 with current state-of-art loss functions in Table 5. We note that these current best-performing methods [4, 17, 34, 39] are based on multi-class classification and belong to the triplet learning paradigm. In contrast, our SphereFace2 is the only method that is based on binary classification and adopts the pair-wise learning paradigm. Following [16], we re-implement SphereFace [17] with hard feature normalization for fair comparison. We observe that

| Loss Function | LFW | AgeDB-30 | CA-LFW | CP-LFW | Combined |
|---|---|---|---|---|---|
| Softmax Loss | 98.20 | 87.23 | 88.17 | 84.85 | 89.05 |
| Coco Loss [20] | 99.16 | 90.23 | 91.47 | 89.53 | 92.4 |
| SphereFace [16, 17] | **99.55** | 92.88 | 92.55 | 90.90 | 93.75 |
| CosFace [39] | 99.51 | 92.98 | 92.83 | 91.03 | 93.89 |
| ArcFace [4] | 99.47 | 91.97 | 92.47 | 90.85 | 93.97 |
| Circle Loss [34] | 99.48 | 92.23 | 92.90 | **91.17** | 93.78 |
| CurricularFace [10] | 99.53 | 92.47 | 92.90 | 90.65 | 93.70 |
| SphereFace2 | 99.50 | **93.68** | **93.47** | 91.07 | **94.28** |

Table 5: Comparison of different loss functions. We take the released source code of these methods and carefully tune the hyperparameters to achieve optimal performance. Results are in % and higher values are better.

the verification accuracies on LFW are saturated around 99.5%. On both AgeDB-30 and CA-LFW datasets, SphereFace2 achieves the best accuracy, outperforming the second best results by a significant margin. The results on the combined datasets also validate the effectiveness of SphereFace2.

## 3.2 EVALUATIONS ON LARGE-SCALE BENCHMARKS

We use three challenging face recognition benchmarks, IJB-B, IJB-C, and MegaFace to evaluate SphereFace2 (with $\lambda = 0.7, r = 40, m = 0.4, t = 3$). We use 64-layer CNNs for all the methods here.

**IJB datasets**. IJB-B [44] has 21.8K still images (including 11.8K faces and 10k non-faces) and 55K frames from 7K videos. The total number of identities is 1,845. We follow the standard 1:1 verification and 1:N identification protocols for experiments. The protocol defines 12,115 templates, where each template consists of multiple images and/or frames. Matching is performed based on the defined templates. Specifically, 10,270 genuine matches and 8M impostor matches are constructed in 1:1 verification protocol, and 10,270 probes and 1,875 galleries are

| | 1:1 Veri. TAR @ FAR | | | 1:N Iden. TPIR @ FPIR | | |
|---|---|---|---|---|---|---|
| Method | 1e-5 | 1e-4 | 1e-3 | rank-1 | 1e-2 | 1e-1 |
| VGGFace2 (SENet) [2] | 67.1 | 80.0 | 88.8 | 90.1 | 70.6 | 83.9 |
| MN-vc [47] | - | 83.1 | 90.9 | - | - | - |
| Comparator Nets [46] | - | 84.9 | 93.7 | - | - | - |
| SphereFace | 80.75 | 89.41 | 94.18 | **93.49** | 73.49 | 87.70 |
| CosFace | 79.62 | 88.61 | 94.10 | 92.90 | 73.80 | 86.89 |
| ArcFace | 80.59 | 89.11 | 94.25 | 93.23 | 74.81 | 87.28 |
| Circle Loss | 78.34 | 88.56 | 94.12 | 92.54 | 72.54 | 86.46 |
| SphereFace2 | **85.40** | **91.31** | **94.80** | 93.32 | **76.89** | **89.91** |

Table 6: Results on IJB-B. We cite the results from the original papers for [2, 46, 47]. For the re-implemented methods, we use the hyperparameters that lead to the best results on the validation set. Results are in % and higher values are better.

constructed in 1:N identification protocol. IJB-C is an extension of IJB-B, comprising 3,531 identities with 31.3K still images and 117.5K frames from 11.8K videos. The evaluation protocols of IJB-C are similar to IJB-B. The details of the protocols are summarized in Appendix. We report the true accept rates (TAR) at different false accept rates (FAR) for verification, and true positive identification rates (TPIR) at different false positive identification rates (FPIR) for identification, as shown in Table 6.

We make several observations based on the evaluation results of IJB-B (Table 6). First, SphereFace2 produces significant improvements over other state-of-the-art methods, especially at low FARs and FPIRs. Specifically, SphereFace2 outperforms CosFace by 5.37% at FAR=1e-5 and 2.70% at FAR=1e-4 in 1:1 verification, 3.09% at FPIR=1e-2 and 3.02% at FPIR=1e-1 in 1:N identification. These significant performance gains suggest

| Method | 1:1 Veri. TAR @ FAR | | | 1:N Iden. TPIR @ FPIR | | |
|---|---|---|---|---|---|---|
| | 1e-5 | 1e-4 | 1e-3 | rank-1 | 1e-2 | 1e-1 |
| VGGFace2 (SENet) [2] | 74.7 | 84.0 | 91.0 | 91.2 | 74.6 | 84.2 |
| MN-vc [47] | - | 86.2 | 92.7 | - | - | - |
| Comparator Nets [46] | - | 88.5 | 94.7 | - | - | - |
| SphereFace | 86.07 | 91.96 | 95.66 | **94.83** | 83.15 | 89.45 |
| CosFace | 85.13 | 90.98 | 95.47 | 94.25 | 80.89 | 88.17 |
| ArcFace | 86.12 | 91.60 | 95.72 | 94.71 | 82.21 | 88.94 |
| Circle Loss | 84.05 | 90.83 | 95.44 | 93.64 | 80.07 | 87.61 |
| SphereFace2 | **89.04** | **93.25** | **96.03** | 94.70 | **85.65** | **91.08** |

Table 7: Results on IJB-C. The testing instances of IJB-C are twice as many as those in IJB-B. Results are in % and higher is better.

that the pair-wise learning paradigm is very useful in improving the robustness of a face recognition system. Similar observations can also be found in the results of IJB-C (Table 7) and the ROC curves results (Fig. 9). Second, the performance is getting saturated for verification rate at FAR=1e-3. Compared to other methods, SphereFace2 can still improve the results by 0.67% - 1.02% on IJB-B and 0.43% - 1.04% on IJB-C. Third, the rank-1 identification performance of SphereFace2 is slightly better than CosFace, ArcFace, Circle Loss, and comparable to SphereFace. Overall, SphereFace2 performs significantly better than current best-performing methods on these two challenging datasets.

**MegaFace**. We further evaluate SphereFace2 on the MegaFace dataset. This is a challenging testing benchmark to evaluate the performance of face recognition methods at the million scale of distractors. It contains a gallery set with more than 1 million images from 690K different identities, and a probe set with 3,530 images from 530 identities. MegaFace provides two testing protocols for identification and verification. We evaluate on both and

| | Iden. ($10^6$ Distractors) | | Veri. (FPR=1e-6) | |
|---|---|---|---|---|
| label refined | ✗ | ✓ | ✗ | ✓ |
| SphereFace | 71.53 | 86.49 | 85.02 | 88.48 |
| CosFace | 71.65 | 86.21 | 85.45 | 88.35 |
| ArcFace | 73.65 | 87.88 | 87.77 | 89.88 |
| Circle Loss | 71.32 | 85.94 | 84.34 | 87.96 |
| SphereFace2 | **74.38** | **89.84** | **89.19** | **91.94** |

Table 8: Results on MegaFace. Because of mislabeled samples in MegaFace, we present the results before and after label refinement.

report the results in Table 8. The gains are consistent with the IJB datasets. Under the same training setup, SphereFace2 outperforms current state-of-the-art methods by a large margin.

## 3.3 NOISY LABEL LEARNING

Since the pair-wise labels used in SphereFace2 provide weaker supervision than the class-wise labels, we perform experiments to evaluate the robustness of SphereFace2 in label-noisy training. We randomly alter 20%, 40%, 60% and 80% of the labels for each class. The four noisy datasets are used to train CosFace, ArcFace, and

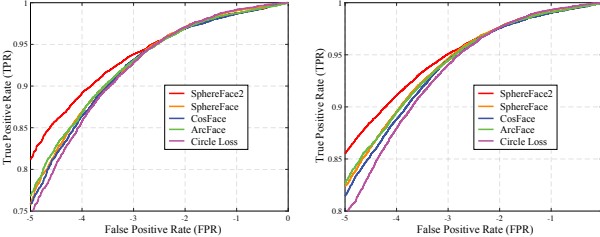

Figure 9: The ROC curves of SphereFace2 and other start-of-art methods on IJB-B (left) and IJB-C (right) datasets.

SphereFace2 separately. We evaluate the trained models on the combined validation set. From Fig 10 (left), SphereFace2 shows stronger robustness to noisy labels than CosFace and ArcFace, as its performance degrades significantly more slowly as the ratio of noisy labels increases from 0 to 0.8.

## 3.4 MODEL PARALLELIZATION

We follow [4] to parallelize the loss computations for CosFace, ArcFace, and SphereFace2. These methods are trained with 1 million identities. Fig. 10 (right) shows how the number of processed images per second changes with different numbers of GPUs. Note that we do not include the feature extraction here. CosFace and ArcFace have negli-

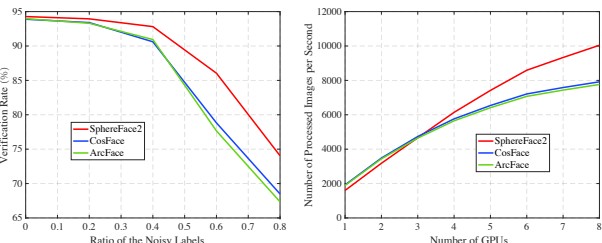

Figure 10: Left: evaluations on the robustness to noisy labels. Right: evaluations on the multi-GPU model parallelization.

gible difference on running time. When a single GPU is used (*i.e.*, no model parallelization), CosFace and ArcFace are slightly faster than SphereFace2. As the number of GPUs increases, the acceleration of SphereFace2 is more significant, due to less communication cost over GPUs. The near linear acceleration for SphereFace2 is owing to its proxy-based pair-wise formulation.

## ACKNOWLEDGEMENTS

AW acknowledges support from a Turing AI Fellowship under grant EP/V025379/1, The Alan Turing Institute, and the Leverhulme Trust via CFI. RS is partially supported by the Defence Science and Technology Agency (DSTA), Singapore under contract number A025959, and this paper does not reflect the position or policy of DSTA and no official endorsement should be inferred.

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

# Appendix

## A  STATISTICS FOR THE USED DATASETS

| Dataset | # of ID | # of img | split |
|---|---|---|---|
| VGGFace2 [2] | 8.6K | 3.1M | train |
| LFW [9] | 5,749 | 13,233 | val |
| Age-DB30 [24] | 568 | 16,488 | val |
| CA-LFW [49] | 5,749 | 11,652 | val |
| CP-LFW [50] | 5,749 | 12,174 | val |
| IJB-B [44] | 1,845 | 76.8K | test |
| IJB-C [21] | 3,531 | 148.8K | test |
| MegaFace (pro.) [11] | 530 | 3,530 | test |
| MegaFace (dis.) [22] | 690K | 1M | test |

Table 9: Statistics for the used datasets.

## B  STATISTICS OF IJB TEST PROTOCOLS

| | | IJB-B [44] | IJB-C [21] |
|---|---|---|---|
| | # of templates | 12,115 | 23,124 |
| 1:1 Verification | # of genuine matches | 10,270 | 19,557 |
| | # of imposter matches | 8M | 15M |
| 1:N Identification | # of probes | 10,270 | 19,593 |
| | # of galleries | 1,875 | 3,531 |

Table 10: The evaluation statistics of the IJB datasets.

## C  MORE HYPERPARAMETER EXPERIMENTS

We additionally provide the results under different hyperparameters. We follow the same settings as in the main paper by training a 20-layer CNN [17] on VGGFace2. First, we vary the hyperparameter $r$ from 20 to 50 and the results in Table 11 show that both $r = 30$ and $r = 40$ work reasonably well. Second, we vary the hyperparameter $m$ from 0.2 to 0.5 and evaluate how the size of margin affects the performance. The results in Table 11 show that $m = 0.3$ generally achieves the best performance. Last, we evaluate how the hyperparameter $t$ in similarity adjustment may affect the performance. Table 11 shows that similarity adjustment is generally helpful for performance (*i.e.*, $t = 2, 3, 4, 5$ works better than $t = 1$) and $t = 3$ achieves the best combined performance.

| $\lambda$ | $r$ | $m$ | $t$ | LFW | AgeDB-30 | CA-LFW | CP-LFW | Combined |
|---|---|---|---|---|---|---|---|---|
| 0.7 | 20 | 0.4 | 3 | 99.56 | 92.97 | 92.85 | 90.97 | 93.96 |
| 0.7 | 30 | 0.4 | 3 | **99.62** | 93.22 | 93.35 | 91.02 | 94.28 |
| 0.7 | 40 | 0.4 | 3 | 99.55 | **93.92** | **93.73** | 90.98 | **94.34** |
| 0.7 | 50 | 0.4 | 3 | 99.48 | 93.42 | 93.53 | **91.07** | 94.22 |
| 0.7 | 30 | 0.2 | 3 | 99.47 | 92.45 | 93.40 | 90.80 | 93.93 |
| 0.7 | 30 | 0.3 | 3 | 99.53 | 93.47 | **93.83** | **91.58** | **94.52** |
| 0.7 | 30 | 0.4 | 3 | 99.50 | **93.68** | 93.47 | 91.07 | 94.28 |
| 0.7 | 30 | 0.5 | 3 | **99.57** | 92.82 | 92.97 | 90.82 | 93.38 |
| 0.7 | 30 | 0.4 | 0.3 | 98.83 | 86.03 | 87.48 | 86.21 | 89.00 |
| 0.7 | 30 | 0.4 | 0.5 | 99.06 | 86.46 | 88.88 | 87.20 | 89.57 |
| 0.7 | 30 | 0.4 | 0.7 | 99.60 | 92.36 | 93.11 | 90.96 | 93.57 |
| 0.7 | 30 | 0.4 | 1 | 99.58 | 92.63 | 93.30 | 90.33 | 93.73 |
| 0.7 | 30 | 0.4 | 2 | **99.62** | 93.22 | 93.35 | 91.02 | 94.05 |
| 0.7 | 30 | 0.4 | 3 | 99.50 | **93.68** | 93.47 | 91.07 | **94.28** |
| 0.7 | 30 | 0.4 | 4 | 99.50 | 93.58 | **93.48** | 90.98 | 94.17 |
| 0.7 | 30 | 0.4 | 5 | 99.58 | 92.90 | 93.45 | **91.12** | 94.16 |

Table 11: Ablations on parameter $r$, $\gamma$, and $t$. Results are in % and higher values are better.

## D  SIMILARITY ADJUSTMENT FOR OTHER METHODS

To empirically show the comparison between SphereFace2 and other methods with SA, we present the experiments on IJBB and IJBC datasets. As shown in Table 12, similarity adjustment works well for SphereFace2, and applying it to multi-class classification losses is not as useful as in SphereFace2.

| Methods on IJBB | 1:1 Veri. TAR @ FAR | | | 1:N Iden. TPIR @ FPIR | | | Methods on IJBC | 1:1 Veri. TAR @ FAR | | | 1:N Iden. TPIR @ FPIR | | |
|---|---|---|---|---|---|---|---|---|---|---|---|---|---|
| | 1e-5 | 1e-4 | 1e-3 | rank-1 | 1e-2 | 1e-1 | | 1e-5 | 1e-4 | 1e-3 | rank-1 | 1e-2 | 1e-1 |
| CosFace w/o SA | 79.35 | 88.05 | 93.71 | 92.54 | 72.20 | 86.40 | CosFace w/o SA | 84.20 | 90.53 | 95.12 | 93.73 | 80.04 | 87.46 |
| CosFace w/ SA | 75.72 | 86.57 | 93.07 | 92.59 | 68.48 | 84.38 | CosFace w/ SA | 82.92 | 89.83 | 94.54 | 93.77 | 78.46 | 86.26 |
| ArcFace w/o SA | 78.12 | 87.11 | 93.29 | 92.18 | 69.86 | 84.88 | ArcFace w/o SA | 82.54 | 89.53 | 94.79 | 93.20 | 78.35 | 86.09 |
| ArcFace w/ SA | 78.88 | 88.06 | 93.71 | 92.65 | 71.62 | 86.08 | ArcFace w/ SA | 84.62 | 90.53 | 95.12 | **93.78** | 80.52 | 87.55 |
| SphereFace2 | **81.11** | **89.05** | **94.11** | **92.66** | **73.54** | **87.32** | SphereFace2 | **85.38** | **91.32** | **95.24** | 93.74 | **81.98** | **88.54** |

Table 12: Comparison of SphereFace2, CosFace and ArcFace (with or without similarity adjustment).

## E  GRADIENT COMPARISON

We compare the gradient between the multi-class softmax-based loss and SphereFace2 in Table 13. We observe that the gradient updates in SphereFace2 can be performed with only the corresponding classifier weights (*i.e.*, no summation over classifier weights of different classes). Therefore, the classifier layer in the SphereFace2 framework can be back-propagated with the local GPU and involve no communication overhead.

| | Multi-class Softmax-based Loss | SphereFace2 |
|---|---|---|
| $L$ | $-\log\left(\frac{\exp(\boldsymbol{W}_y^\top \boldsymbol{x})}{\sum_i \exp(\boldsymbol{W}_i^\top \boldsymbol{x})}\right)$ | $\log\left(1+\exp(-\boldsymbol{W}_y^\top \boldsymbol{x})\right)+\sum_{i\neq y}\log\left(1+\exp(\boldsymbol{W}_i^\top \boldsymbol{x})\right)$ |
| $\frac{\partial L}{\partial \boldsymbol{W}_i}$ $(i\neq y)$ | $\frac{\exp(\boldsymbol{W}_i^\top \boldsymbol{x})}{\sum_j \exp(\boldsymbol{W}_j^\top \boldsymbol{x})}\cdot\boldsymbol{x}$ | $\frac{\exp(\boldsymbol{W}_i^\top \boldsymbol{x})}{1+\exp(\boldsymbol{W}_i^\top \boldsymbol{x})}\cdot\boldsymbol{x}$ |
| $\frac{\partial L}{\partial \boldsymbol{W}_y}$ | $\left(\sum_{i\neq y}\frac{\exp(\boldsymbol{W}_i^\top \boldsymbol{x})}{\sum_j \exp(\boldsymbol{W}_j^\top \boldsymbol{x})}-1\right)\cdot\boldsymbol{x}$ | $-\frac{\exp(\boldsymbol{W}_y^\top \boldsymbol{x})}{1+\exp(\boldsymbol{W}_y^\top \boldsymbol{x})}\cdot\boldsymbol{x}$ |

Table 13: Gradient comparisons between the multi-class softmax-based loss and SphereFace2. Here we omit the constant terms, *e.g.* bias, margin, etc., since they do not affect the conclusion.

## F  DIFFERENT FORMS OF ANGULAR MARGIN IN SPHEREFACE2

Although we use the additive margin as an example in the main paper, it is natural to consider the other forms of angular margin in SphereFace2. Specifically, we first revisit the final loss function that uses a particular type of additive margin [37, 39] (also used as the example in the main paper):

$$L_{\text{SF2-C}} = \frac{\lambda}{r}\cdot\log\left(1+\exp\left(-r\cdot(g(\cos(\theta_y))-m)-b\right)\right)+\frac{1-\lambda}{r}\cdot\sum_{i\neq y}\log\left(1+\exp\left(r\cdot(g(\cos(\theta_i))+m)+b\right)\right).$$

For another type of additive margin [4] and the multiplicative margin [16, 17], we implement them using the Characteristic Gradient Detachment (CGD) trick [16] to enable stable training. Therefore, we use the following loss function to implement the ArcFace-type additive margin in SphereFace2:

$$L_{\text{SF2-A}} = \frac{\lambda}{r}\cdot\log\left(1+\exp\left(-r\cdot g(\cos(\theta_y))-r\cdot\text{Detach}\left(g\left(\cos\left(\min\left(\pi,\theta_y+m\right)\right)\right)-g(\cos(\theta_y))\right)-b\right)\right)$$
$$+\frac{1-\lambda}{r}\cdot\sum_{i\neq y}\log\left(1+\exp\left(r\cdot g(\cos(\theta_i))+b\right)\right),$$

where $\text{Detach}(\cdot)$ is a gradient detachment operator that stops the back-propagated gradients. For details of how CGD works, refer to [16]. For the multiplicative margin, we adopt an improved version from SphereFace-R [16] (*i.e.*, SphereFace-R v1), which yields

$$L_{\text{SF2-M}} = \frac{\lambda}{r}\cdot\log\left(1+\exp\left(-r\cdot g(\cos(\theta_y))-r\cdot\text{Detach}\left(g\left(\cos(\min(m,\frac{\pi}{\theta_y})\cdot\theta_y)\right)-g(\cos(\theta_y))\right)-b\right)\right)$$
$$+\frac{1-\lambda}{r}\cdot\sum_{i\neq y}\log\left(1+\exp\left(r\cdot g(\cos(\theta_i))+b\right)\right).$$

For both ArcFace-type and multiplicative margin, we do not inject the angular margin to the negative samples. Considering the two cases with or without angular margin for the negative samples, we note that a learnable bias $b$ makes both cases equivalent. The only difference is that the optimal choice for the margin parameter $m$ may vary for the two cases.

Then we conduct experiments to empirically compare them. We adopt the same training settings as Table 5 (*i.e.*, SFNet-20 [16, 17] without batch normalization). The results are given in Table 14, Table 15, and Table 16. Here we term SphereFace2 with CosFace-type additive margin as **SphereFace2-C**, SphereFace2 with ArcFace-type additive margin as **SphereFace2-A** and SphereFace2 with multiplicative additive margin as **SphereFace2-M**. The margins for SphereFace2-A and SphereFace2-M are 0.5 and 1.7, respectively. We observe that different types of angular margin perform similarly in general. Note that we did not carefully tune the hyperparameters for SphereFace2-A and Sphereface2-M and the performance is already very competitive. We believe that the performance can be further improved by a more systematic hyperparameter search.

| Loss Function | LFW | AgeDB-30 | CA-LFW | CP-LFW | Combined |
|---|---|---|---|---|---|
| Softmax Loss | 98.20 | 87.23 | 88.17 | 84.85 | 89.05 |
| Coco Loss [20] | 99.16 | 90.23 | 91.47 | 89.53 | 92.4 |
| SphereFace [16, 17] | 99.55 | 92.88 | 92.55 | 90.90 | 93.75 |
| CosFace [39] | 99.51 | 92.98 | 92.83 | 91.03 | 93.89 |
| ArcFace [4] | 99.47 | 91.97 | 92.47 | 90.85 | 93.97 |
| Circle Loss [34] | 99.48 | 92.23 | 92.90 | **91.17** | 93.78 |
| CurricularFace [10] | 99.53 | 92.47 | 92.90 | 90.65 | 93.70 |
| SphereFace2-C | 99.50 | **93.68** | 93.47 | 91.07 | **94.28** |
| SphereFace2-A | 99.51 | 93.53 | **93.75** | 91.01 | 94.19 |
| SphereFace2-M | **99.58** | 93.63 | 93.66 | 90.95 | 94.19 |

Table 14: Comparison of different loss functions. Results are in % and higher values are better.

| Methods on IJBB | 1:1 Veri. TAR @ FAR | | | 1:N Iden. TPIR @ FPIR | | |
|---|---|---|---|---|---|---|
| | 1e-5 | 1e-4 | 1e-3 | rank-1 | 1e-2 | 1e-1 |
| CosFace | 79.35 | 88.05 | 93.71 | 92.54 | 72.20 | 86.40 |
| ArcFace | 78.12 | 87.11 | 93.29 | 92.18 | 69.86 | 84.88 |
| SphereFace2-C | **82.36** | **89.54** | 93.94 | 92.61 | 72.52 | **88.22** |
| SphereFace2-A | 80.89 | **89.54** | **94.10** | **92.68** | 72.87 | 87.96 |
| SphereFace2-M | 81.20 | 89.29 | **94.10** | 92.53 | **74.52** | 87.95 |

Table 15: Comparison of CosFace, ArcFace, and SphereFace2 with different margin types on IJB-B dataset.

| Methods on IJBC | 1:1 Veri. TAR @ FAR | | | 1:N Iden. TPIR @ FPIR | | |
|---|---|---|---|---|---|---|
| | 1e-5 | 1e-4 | 1e-3 | rank-1 | 1e-2 | 1e-1 |
| CosFace | 84.20 | 90.53 | 95.12 | 93.73 | 80.04 | 87.46 |
| ArcFace | 82.54 | 89.53 | 94.79 | 93.20 | 78.35 | 86.09 |
| SphereFace2-C | **86.78** | **91.78** | 95.26 | **93.77** | **83.20** | **89.36** |
| SphereFace2-A | 86.30 | 91.68 | 95.33 | 93.72 | 82.82 | 89.24 |
| SphereFace2-M | 86.38 | 91.61 | **95.38** | 93.72 | 82.71 | 89.29 |

Table 16: Comparison of CosFace, ArcFace, and SphereFace2 with different margin types on IJB-C dataset.

