# OpenReview forum: "SphereFace2: Binary Classification is All You Need for Deep Face Recognition"
_ICLR.cc/2022/Conference — ICLR 2022 Spotlight_

### Official Review · Reviewer_6DJe · 2021-10-27

**Correctness:** 3
**Technical Novelty And Significance:** 3
**Empirical Novelty And Significance:** 4
**Recommendation:** 8
**Confidence:** 3

**Main Review:**

Pros:
- Different from the commonly used multi-class classification, the proposed method uses binary classification to reduce the gap between training and open-set testing, which addresses the face recognition problem from a new perspective.
- The principles for the designed loss function are well-motivated and clearly explained. The geometric interpretations also help to better understand.
- The experiments are comprehensive, and show the state-of-the-art performance of the proposed method.
- The paper is well-written and easy to follow.

Cons:
- In Sec 3.1, the hyperparameters are analyzed with different values. In Sec 3.2, since the backbone network has changed, which hyperparameter setting is adopted? It would be better to clarify this.
- Some results in the text seem to be inconsistent with those in the tables. For examples, in p .8 l. 3, 93.73% (93.87% in Table 2); in p. 8 l. 7-9, 5.37%, 2.74%, 3.50%, and 3.52% (5.78%, 2.70%, 3.09%, and 3.02% in Table 6).
- A few typos, e.g., p. 9 l. 13, Table 6 -> Table 7.


**Summary Of The Paper:**

This paper presents a binary classification training framework, SphereFace2, for face recognition. Instead of multi-class classification, the proposed method adopts pair-wise comparisons learned through a well-designed loss function, which is more suitable for open-set evaluation. Comprehensive experiments demonstrate the effectiveness of the proposed method.

**Summary Of The Review:**

The idea of using binary classification learning to cope with face recognition is well-motivated and reasonable. The framework and loss function design are also clearly explained and further demonstrated by experiments on benchmark datasets. Although there may be some minor errors, I think this is a good paper, and would vote for accept.

---

> ### Author Response · Authors · 2021-11-22
> **Response to reviewer 6DJe**
>
> We sincerely thank the reviewer for recognizing our contribution and novelty! We take every comment seriously and address every concern point by point. We hope that our response resolves your concerns. Any follow-up questions are welcome.
>
> ---
> ---
>
> **Question 1. In Sec 3.1, the hyperparameters are analyzed with different values. In Sec 3.2, since the backbone network has changed, which hyperparameter setting is adopted? It would be better to clarify this.**
>
> **Response 1**. Thanks for the great suggestion. We use $\lambda=0.7, r=40, m=0.4, t=3$. We have added the detailed experimental settings to revision.
>
> ---
>
> **Question 2. Some results in the text seem to be inconsistent with those in the tables. For examples, in p .8 l. 3, 93.73% (93.87% in Table 2); in p. 8 l. 7-9, 5.37%, 2.74%, 3.50%, and 3.52% (5.78%, 2.70%, 3.09%, and 3.02% in Table 6).**
>
> **Response 2**. Sorry for the typos and thanks for pointing them out. We have fixed them in the updated version.
>
> ---
>
> **Question 3. A few typos, e.g., p. 9 l. 13, Table 6 -> Table 7.**
>
> **Response 3**. Thanks for pointing out these typos. We have fixed it in the updated version.

---

> > ### Comment · Reviewer_6DJe · 2021-11-24
> > **My concerns have been addressed**
> >
> > My concerns have been addressed in the revised paper. Thank the authors for their efforts in the revision.
> >
> > Best regards,

---

### Official Review · Reviewer_iYQo · 2021-10-29

**Correctness:** 4
**Technical Novelty And Significance:** 3
**Empirical Novelty And Significance:** 3
**Recommendation:** 8
**Confidence:** 4

**Details Of Ethics Concerns:**

No ethical concern.

**Main Review:**

** Strengths
- The discussion on the advantages of the multi-binary classification strategy is interesting and persuasive. I like the comparison between triplet-based learning and pair-based learning in Figure 2.
- Since the naive solution does not work, the authors proposed some principles in designing an effective loss function. These principles are effective in designing a good loss function, as shown in Table 2.
- The authors did a good job on explaining the proposed components via geometric interpretation.
- SphereFace2 outperforms the state-of-the-art face recognition models on 1:1 verification and 1:N open-set evaluation benchmarks.
- It also allows model parallelization that effectively speeds up the training process when multiple GPUs are available.

** Weaknesses
- The Positive/negative sample balance technique is a bit simple. In fact, the authors manually tuned the hyper-parameter \lambda to get a desirable results while the simplest configuration does not work well (the first row in Table 2). There are existing techniques to handle data imbalance such as focal loss. It is better to have a comparison to these available options.
- The authors did not provide the final values of the hyper-parameters (λ, r, m, and t)
- SphereFace2 performs worse than SphereFace in 1-N rank-1 Indentification tests
- In Section 3.2 and 3.3, the authors used CosFace as the baseline for comparison. ArcFace is a better option.

**Summary Of The Paper:**

This paper proposes a novel face recognition model called SphereFace2. The key idea is to replace the softmax-based multi-class training with multiple binary classification training, which is more consistent with the testing phrase and generalized to both closed/open-set evaluations. To make this novel strategy works, the authors defined some principles in designing an effective loss function, including (1) Positive/negative sample balance, (2) Easy/hard sample mining, (3) Angular margin, and (4) Similarity adjustment. SphereFace2 outperforms the state-of-the-art face recognition models on 1:1 verification and 1:N open-set evaluation benchmarks. It also allows model parallelization that effectively speeds up the training process when multiple GPUs are available.

**Summary Of The Review:**

Overall, this paper presents a novel face recognition model, called SphereFace2, with interesting insights and persuasive explanations. The proposed model outperforms the baselines on popular 1-1 verification and 1-N open-set evaluation. Despite few minor issues, I lean towards accepting this paper.

---

> ### Author Response · Authors · 2021-11-22
> **Response to reviewer iYQo**
>
> We sincerely thank the reviewer for recognizing our contribution and novelty! We take every comment seriously and address every concern point by point. We hope that our response resolves your concerns. Any follow-up questions are welcome.
>
> ---
> ---
>
> **Question 1. The Positive/negative sample balance technique is a bit simple. In fact, the authors manually tuned the hyper-parameter \lambda to get a desirable results while the simplest configuration does not work well (the first row in Table 2). There are existing techniques to handle data imbalance such as focal loss. It is better to have a comparison to these available options.**
>
> **Response 1**. Thanks for the great suggestion. We totally agree with the reviewer that advanced positive/negative sample balance methods will benefit the final performance. We have no doubts that there could be many potentially better choices for each design principle. This paper instead seeks to demonstrate these design principles based on a minimalist ideology. Our future endeavor is to further explore possible design choices for each principle in the SphereFace2 framework.
>
> We also would like to emphasize that the contribution of SphereFace2 is more about the binary classification framework and the corresponding principles to make it work in practice (rather than exhaustively trying every possible design choice). For example, we adopt a simple yet effective way to balance positive/negative samples with a hyperparameter (tuned on validation set). In fact, focal loss also uses a hyperparameter to control the positive/negative sample balance and tunes it based on a validation set, which is the same as our method.
>
> ---
>
> **Question 2. The authors did not provide the final values of the hyper-parameters (λ, r, m, and t)**
>
> **Response 2**. Sorry for missing this. For IJB-B/C and MegaFace, we use the same set of hyperparameters that achieves the best performance on the validation set (LFW, AgeDB-30, CA-LFW, and CP-LFW). Specifically, we use the following hyperparameters: $\lambda=0.7, r=40, m=0.4, t=3$. We have added the final hyperparameters at the beginning of Section 3.2.
>
> ---
>
> **Question 3. SphereFace2 performs worse than SphereFace in 1-N rank-1 Indentification tests**
>
> **Response 3**. On this particular metric, SphereFace2 performs just slightly worse than SphereFace (by 0.13%). However, considering the overall performance on the rest of these metrics, it is safe to say that SphereFace2 is the best one among the compared methods.
>
> ---
>
> **Question 4. In Section 3.2 and 3.3, the authors used CosFace as the baseline for comparison. ArcFace is a better option.**
>
> **Response 4**. Thanks for the great suggestion. Maybe the reviewer means Section 3.3 and 3.4 instead of 3.2 and 3.3 (since we already have ArcFace in Section 3.2)? We use CosFace as the baseline, because the angular margin in SphereFace2 is similar to CosFace (additive margin to $cos(\theta)$, rather than $\theta$). Nonetheless, we thought ArcFace is also an important baseline to compare with. Following the reviewer’s suggestion, we add ArcFace to the comparison as well. Please see the updated paper. We have added the comparison to ArcFace in Section 3.3 and 3.4 to the revision.

---

> > ### Comment · Reviewer_iYQo · 2021-11-23
> > **Your PDF file has not been updated**
> >
> > Dear Authors,
> >
> > I have checked your PDF file, but it does not seem to be updated. I cannot find your final hyperparameters at the beginning of Section 3.2 and ArcFace plots in Figure 10. Could you double-check?
> >
> > Best,

---

> > > ### Author Response · Authors · 2021-11-23
> > > **New version uploaded!**
> > >
> > > Dear Reviewer iYQo,
> > >
> > > Thanks a lot for the reminder! We have updated the paper now. Please see the latest version and let us know if there is any problem.
> > >
> > > Best,

---

> > > > ### Comment · Reviewer_iYQo · 2021-11-24
> > > > **The updated PDF looks good**
> > > >
> > > > The PDF was updated. It looks good to me.
> > > >
> > > > Best regards,

---

### Official Review · Reviewer_4dBH · 2021-11-03

**Correctness:** 3
**Technical Novelty And Significance:** 3
**Empirical Novelty And Significance:** 3
**Recommendation:** 8
**Confidence:** 5

**Main Review:**

### Strengths:
1. The paper is well written with nice presentation and clear description.
2. The proposed method has sound motivation, i.e., bridging the gap between the training process and the actual test protocol when deployed for open-set applications. The effectiveness of the key components are well illustrated from both intuitive and empirical perspectives with figures and statistics.
3. The new loss formulation also benefits parallel training setup which are widely adopted nowadays.
4. Extensive experiments have been conducted to ascertain the final outstanding performance of the proposed method.

### Weaknesses:
1. The beginning of this paper may be misleading. The problem of multi-class softmax-based training when switched to the open-set deployment has been noticed in many early works, such as [25] [27], where contrastive loss and triplet loss were used to address this problem. In this paper, however, the Abstract, Paragraph 1 and the first sentence of Paragraph 2 did not mention this fact and therefore may mislead readers to believe that this work is the first to address this issue.
2. Some statements are vague or unclear, such as:
    * The authors claim that softmax loss introduces competition among different classes. What is the definition of "competition"? Even using the proposed loss function, the feature embedding still receives gradients from all the classes (1 positive and N-1 negative classes) in one training iteration, could that also be a kink of inter-class competition?
    * The categorization of proxy-free and proxy based is unclear. Why is using softmax loss and its variants considered proxy-based (like [28]) but using combination of softmax and contrastive loss is considered proxy-free (like [27])?
    * I didn't understand the example given for similarity adjustment, where the text says "if g(cos(theta)) is ... range of [-0.2, 1], we can end up with cos(theta) ... becoms [-0.9, 1]." But looking at the formulation and Figure 6, it seems that when t > 1, cos(theta) is within [-0.2, 1] while g(cos(theta)) is within [-0.9, 1]. I am not sure if I misunderstood anything and expect more details from the authors feedback.
    * In the end, how is the verification threshold is determined? Based on the validation set or is there a theoretical value? If the threshold is determined from the results on the validation set, how to understand the claim that the proposed method eases the difficulty of choosing a verification threshold compared to other methods?
3. In addition to new formulation of the binary classification training protocol, there are multiple components contributing to the final performance, including sample balancing, easy/hard sample mining, angular margin and similarity adjustment. It is hard to judge which component is play a key role to the results. Looking at Table 2, without SA, the performance (93.87%) is on par with other state-of-the-art works such as ArcFace (93.97%). However, SA can also be applied to other loss functions, then what is the actual benefit by using the binary classification loss formulation?

**Summary Of The Paper:**

This paper has proposed SphereFace2, a new face recognition training scheme that applies a set of binary classification losses to each identities in the dataset, to learn better face embeddings. The proposed method has addressed some limitations of existing training schemes, such as discrepancy between the softmax-based loss and the actual open-set face recognition scenarios. The proposed binary classification protocol works together with a series of optimization strategies, such as easy/hard sample mining and similarity adjustment, to achieve superior performance over state-of-the-art methods as shown in the extensive experiments.

**Summary Of The Review:**

This paper has proposed a relatively new idea on training face recognition models by using multiple binary classification losses. The proposed method has convincing motivation and the technical design is reasonable. Experimental results ascertain the superiority of the proposed method as a unified solution. However, some statements of the paper are misleading or vague. Moreover, as the proposed method comprises multiple effective components, it is also unclear that whether the performance improvements come from the new formulation or from one or two incremental components.

---

> ### Author Response · Authors · 2021-11-22
> **Response to reviewer 4dBH (2)**
>
> **Question 5. How is the verification threshold is determined? Based on the validation set or is there a theoretical value? How to understand the claim that the proposed method eases the difficulty of choosing a verification threshold compared to other methods?**
>
> **Response 5**. Sorry for missing this in the experimental details. The verification threshold (used in evaluation) is determined based on the validation set, which is identical to the settings of the other methods, as explained at the beginning of section 3. The consistent testing protocol across different methods guarantees fair comparisons.
>
> We apologize for the unclear presentation that may cause misunderstanding. The claim of "ease the difficulty of choosing a verification threshold" can be understood as follows. In general, the training of SphereFace2 explicitly encourages a universally consistent verification threshold for separating positive and negative pairs, because we adopt a unique bias for all the binary classifications (see Eqn. 5) and the classification is based on a pair and a threshold (see Fig. 2). In contrast, multi-class softmax losses do not encourage a consistent threshold for positive and negative pairs, because the classification is based on comparisons among different pairs.
>
> More specifically, we denote the similarity score of a positive pair as $s_p$, and that of a negative pair as $s_n$. SphereFace2 maximizes $s_p$ and minimizes $s_n$ separately with respect to a consistent threshold, while multiclass softmax-based losses maximize the relative difference $s_p-s_n$.
>
> Finally, the open-set evaluation is also based on inspecting the similarity scores of sample pairs, which is consistent with the training settings of SphereFace2.
>
> We have improved the presentation about the "verification threshold" in the updated version.
>
> ---
>
> **Question 6. In addition to new formulation of the binary classification training protocol, there are multiple components contributing to the final performance. It is hard to judge which component is play a key role to the results.**
>
> **Response 6**. While having multiple advantages in its formulation (i.e., no gap between training and testing, scalability to multi-GPU parallelization and robustness to label noise), the binary classification training framework still lacks some standard characteristics in its loss function. We want to emphasize that the current popular margin-based softmax losses already have sampling balancing, easy/hard sample mining, and angular margin. Therefore, in order to be comparable to the margin-based softmax loss, we first need to incorporate these three characteristics to the binary classification loss. Therefore, these three components cannot be viewed as additional components, and they are an essential part of SphereFace2. We have given a detailed ablation study in Table 2 of the main paper to justify their effectiveness.
>
> Similarity adjustment is an extra component that is inspired by the special design of the binary classification objective. We show that it works well for SphereFace2. We further verify that applying it to multi-class classification losses is not as useful as SphereFace2 (see Q7 for details).
>
> ---
>
> **Question 7. SA can also be applied to other loss functions, then what is the actual benefit by using the binary classification loss formulation?**
>
> **Response 7**. Thanks for the great suggestion on the ablation study. We totally agree with the reviewer that it would be more convincing to empirically compare SphereFace2 and other methods with SA. We use the validation set to choose the best $t$ in SA for each method, and the results on two large-scale testing sets are given as follows. The results show that SA is more beneficial for our binary classification training.
>
>
> |IJBB|1:1 Veri. TAR at FAR|1:N Iden. TPIR at FPIR|
> |---|:---:|:---:|
> |**method**|**1e-5** **$~~~~$1e-4$~~~~$** **1e-3**|**rank-1** **$~~~~$1e-2$~~~~$** **1e-1**|
> |CosFace w/o SA|79.35 $~~~$88.05$~~~$ 93.71|$~~$92.54 $~~~~$72.20$~~~$ 86.40|
> |CosFace w/ SA|75.72 $~~~$86.57$~~~$ 93.07|$~~$92.59 $~~~~$68.48$~~~$ 84.38|
> |ArcFace w/o SA|78.12 $~~~$87.11$~~~$ 93.29|$~~$92.18 $~~~~$69.86$~~~$ 84.88|
> |ArcFace w/ SA|78.88 $~~~$88.06$~~~$ 93.71|$~~$92.65 $~~~~$71.62$~~~$ 86.08|
> |SphereFace2|**81.11** **$~~~$89.05$~~~$** **94.11**|**$~~$92.66** **$~~~~$73.54$~~~$** **87.32**|
>
> |IJBC|1:1 Veri. TAR at FAR|1:N Iden. TPIR at FPIR|
> |---|:---:|:---:|
> |**method**|**1e-5** **$~~~~$1e-4$~~~~$** **1e-3**|**rank-1** **$~~~~$1e-2$~~~~$** **1e-1**|
> |CosFace w/o SA|84.20 $~~~$90.53$~~~$ 95.12|$~~$93.73 $~~~~$80.04$~~~$ 87.46|
> |CosFace w/ SA|82.92 $~~~$89.83$~~~$ 94.54|$~~$93.77 $~~~~$78.46$~~~$ 86.26|
> |ArcFace w/o SA|82.54 $~~~$89.53$~~~$ 94.79|$~~$93.20 $~~~~$78.35$~~~$ 86.09|
> |ArcFace w/ SA|84.62 $~~~$90.53$~~~$ 95.12| **$~~$93.78** $~~~~$80.52$~~~$ 87.55|
> |SphereFace2|**85.38** **$~~~$91.32$~~~$** **95.24**|$~~$93.74 **$~~~~$81.98$~~~$** **88.54**|

---

> > ### Comment · Reviewer_4dBH · 2021-11-29
> > **Most of my concerns have been addressed except for some minor points**
> >
> > The authors have made reasonable rebuttal on the questions I raised in my original review. Still, some minor points remain unclear to me, described in the following:
> > 1. It is still unclear that what the definition of "proxy" is and what role it plays in the training process.
> > 2. It is good to see new results on SA + other methods, but it would be better if the authors provide some insightful analysis on the reason that SA is more suitable for the proposed binary classification training protocol.
> >
> > Overall the updated paper is much better than the first version to me and therefore I am willing to increase my rating.

---

> > > ### Author Response · Authors · 2021-11-30
> > > **Response to the additional questions**
> > >
> > > We sincerely thank the reviewer for carefully reading our paper and rebuttal. We are glad to see that our rebuttal has clarified the reviewer’s concerns. For any additional concerns, we are more than happy to address them.
> > >
> > > ---
> > >
> > > **Question 1. It is still unclear that what the definition of "proxy" is and what role it plays in the training process.**
> > >
> > > **Response 1**. The term “proxy” is first proposed in [a] for deep metric learning, and later adopted by many approaches (such as [b]) in face recognition. A proxy point (or a proxy) refers to a vector that is used for representing a group of embeddings from a subject. The proxy essentially has the same meaning as the term “class center” used in [c] and [d].
> > >
> > > In SphereFace2, we construct a proxy for each class, denoted by $W_j$ (see section 2.1). For a deep feature vector $x$ belonging to the $y$-th class, $W_y$ and $W_j (j \neq y)$ are considered as the positive proxy and negative proxy, respectively.
> > >
> > > We will add the explanation to avoid confusion in the revision.
> > > \
> > > \
> > > [a] Movshovitz-Attias, Yair, et al. "No fuss distance metric learning using proxies." ICCV 2017.\
> > > [b] Sun, Yifan, et al. "Circle loss: A unified perspective of pair similarity optimization." CVPR 2020.\
> > > [c] Wen, Yandong, et al. "A discriminative feature learning approach for deep face recognition." ECCV 2016.\
> > > [d] Deng, Jiankang, et al. "Sub-center arcface: Boosting face recognition by large-scale noisy web faces." ECCV 2020.
> > >
> > > ---
> > >
> > > **Question 2. It is good to see new results on SA + other methods, but it would be better if the authors provide some insightful analysis on the reason that SA is more suitable for the proposed binary classification training protocol.**
> > >
> > > **Response 2**. Thanks for the great suggestion. We provide the analysis using the same notations as response 5.
> > > SphereFace2 optimizes the similarity scores ($s_p$ and $s_n$) individually with respect to a universal threshold (see Eqn. 5). Therefore, SA is beneficial to the training process, because it can balance the distribution of $s_n$ and $s_p$ (see Fig. 5) and increase the effective score range (see Fig. 6).
> > > On the other hand, multi-class softmax losses (such as SphereFace, CosFace, and ArcFace) optimize the relative difference $s_p-s_n$. As a result, applying SA to $s_p$ and $s_n$ has little effect on the resulting $s_p-s_n$, which leads to little improvement for the performance.
> > >
> > > The analysis will be added to the experiment section in the revision.

---

> > > > ### Comment · Reviewer_4dBH · 2021-12-01
> > > > **Clarified**
> > > >
> > > > Thanks for the further explanation. I have no more questions.

---

> ### Author Response · Authors · 2021-11-22
> **Response to reviewer 4dBH (1)**
>
> We sincerely thank the reviewer for the detailed comments. We carefully address each of the reviewer’s concerns below and hope that our response resolves your concerns. Any follow-up questions are welcome.
>
> ---
> ---
>
> **Question 1. The beginning of this paper may be misleading. The problem of multi-class softmax-based training when switched to the open-set deployment has been noticed in many early works, such as [25] [27], where contrastive loss and triplet loss were used to address this problem. In this paper, however, the Abstract, Paragraph 1 and the first sentence of Paragraph 2 did not mention this fact and therefore may mislead readers to believe that this work is the first to address this issue.**
>
> **Response 1**. Thanks for pointing this out. We agree with the reviewer that there are early works that notice the problem of multi-class softmax-based training. We don’t mean to claim that we are the first to notice this problem. The confusion may come from the abstract where we say "In the paper, we first identify the discrepancy … and then discuss the potential limitation ...". We are sorry about the potential confusion caused. In this sentence, we simply mean that we start our paper by identifying the discrepancy between training and evaluation in the existing multi-class classification framework. We have fixed the relevant presentation to avoid any confusion.
>
> We want to emphasize that our major contribution is to propose a new framework to address this discrepancy, rather than identifying the discrepancy between training and testing. Current state-of-the-art methods still try to address this discrepancy under a multi-class (or triplet-based) training framework (e.g., FaceNet, SphereFace, CosFace, ArcFace, etc.), while SphereFace2 adopts a novel pair-wise learning paradigm with proxies.
>
> ---
>
> **Question 2. The authors claim that softmax loss introduces competition among different classes. What is the definition of "competition"? Even using the proposed loss function, the feature embedding still receives gradients from all the classes (1 positive and N-1 negative classes) in one training iteration, could that also be a kink of inter-class competition?**
>
> **Response 2**. The "competition" property is in the output of the softmax function, where increasing the confidence of one entry necessarily results in decreasing the confidence of at least one of the other, since the confidences of all entries are summed to 1. SphereFace2 gets rid of this hard constraint. The output confidences in SphereFace2 are not regularized by each other.
>
> From the gradient perspective, such a competition caused by softmax function leads to the coupling of the gradients w.r.t. the classifier weights. That is to say, to update $\frac{\partial L}{\partial W_i}$ requires the weights from all the classes ($W_{1,...,N}$). In contrast, the gradient updates in SphereFace2 can be performed individually with the weights of the corresponding classifier. See Appendix D for a detailed comparison.
>
> ---
>
> **Question 3. The categorization of proxy-free and proxy based is unclear. Why is using softmax loss and its variants considered proxy-based (like [28]) but using combination of softmax and contrastive loss is considered proxy-free (like [27])?**
>
> **Response 3**. Sorry for the confusion. To address this issue, we have modified the categorization strategy in Table 1 and added explanations to its caption. Specifically, the hybrid loss functions are marked by a special symbol "*" and appear in multiple places in Table 1, depending on what kind of loss they consist of. For example, [27] uses a combination of softmax and contrastive loss, so it is in both "triplet & w/ proxy" and "pair & w/o proxy".
>
> However, we would like to emphasize that this confusion does not affect the novelty of SphereFace2, since we are the only deep FR method that adopts a pair-wise learning paradigm with proxies.
>
> ---
>
> **Question 4. I didn't understand the example given for similarity adjustment, where the text says "if g(cos(theta)) is ... range of [-0.2, 1], we can end up with cos(theta) ... becoms [-0.9, 1]." But looking at the formulation and Figure 6, it seems that when t > 1, cos(theta) is within [-0.2, 1] while g(cos(theta)) is within [-0.9, 1]. I am not sure if I misunderstood anything and expect more details from the authors feedback.**
>
> **Response 4**. Sorry for the inconsistency. It is actually a typo in the text. We have fixed it as follows: "if $cos(\theta)$ is ... range of [-0.2, 1], we can end up with $g(cos(\theta))$ ... becomes [-0.9, 1]."
>
> Beyond this typo, we notice that our ablation for $t$ in similarity adjustment misses the cases where $t<1$. We have conducted the experiment of $t=0.3,0.5,0.7$, and the results have been added to Fig. 11 in Appendix C. The results further verify the effectiveness of the current design of similarity adjustment.

---

### Author Response · Authors · 2021-11-22
**General response from authors to the reviewers and AC**

Dear Reviewers and ACs

We thank all the reviewers and ACs for spending time on our submission. We are deeply appreciative for all these valuable and constructive comments.

First of all, we would like to thank all the reviewers for the consistent recognition of our contributions (especially novelty). We notice that most concerns are about presentation. We try our best to improve the presentation of the paper in this short amount of time, and we will keep polishing our paper. We have conducted all the requested experiments and added the missing experimental details to our paper. We hope that our responses address reviewers’ concerns.

**We mark the added and modified content as Green in the updated paper**

To facilitate future research, we will release our codebase to ensure that every result in our paper is reproducible. If there are any additional concerns or questions, we will be more than happy to address them.

Respectfully,

Authors

---

### Decision · Program_Chairs · 2022-01-20

**Decision:**

Accept (Spotlight)

**Comment:**

All reviewers agree that the proposed SphereFace2 approach - training face recognition models by using multiple binary classification losses - is interesting and innovative. The reviewers agree that the paper is well written and are satisfied with the presented experimental study. The rebuttal addressed all additionally raised questions. I believe that the paper will be of interest to the audience attending ICLR and would recommend a presentation of the work as a spotlight.